# How Does Meaning-Centered Coping Influence College Students’ Mental Health? The Mediating Roles of Interdependent Self-Construal and School Connectedness

**DOI:** 10.3390/bs15070955

**Published:** 2025-07-15

**Authors:** Qin Lu, Qian Chen, Yuanhao Zhang, Zongkui Zhou

**Affiliations:** 1Key Laboratory of Adolescent Cyberpsychology and Behavior (CCNU), Ministry of Education, Wuhan 430079, China; shida16308@163.com (Q.L.);; 2Key Laboratory of Human Development and Mental Health of Hubei Province, School of Psychology, Central China Normal University, Wuhan 430079, China; 3Mental Health Education Centre for Students, Qinhai University, Xining 810016, China; 4Wuhan Police Vocational College, Wuhan 430040, China

**Keywords:** meaning-centered coping, school connectedness, interdependent self-construal, mental health, college students

## Abstract

Meaning-centered coping is regarded as an effective strategy for managing stress and preventing mental disorders. However, it remains unclear how it influences mental health by affecting both the self and social connection dimensions. This study investigated 856 college students through a questionnaire, examining how meaning-centered coping affects their mental health (depression and anxiety). Additionally, this study explored the roles of interdependent self-construal and school connectedness as mediators in this process. The results indicate that meaning-centered coping influences mental health either via the mediation of school connectedness alone (indirect effect for depression: β = −0.08, 95% CI [−0.11, −0.04]; for anxiety: β = −0.06, 95% CI [−0.10, −0.03]) or via the sequential mediation of interdependent self-construal and school connectedness (indirect effect for depression: β = −0.08, 95% CI [−0.11, −0.05]; for anxiety: β = −0.06, 95% CI [−0.10, −0.04]). This study reveals that college students who are skilled at seeking and reconstructing their sense of meaning can effectively cope with stress and alleviate related depression and anxiety. This coping mechanism operates through perceived school connectedness or through activated interdependent self-construal followed by perceived school connectedness, subsequently reducing anxiety and depression induced by chronic stress. This study theoretically deepens the comprehension of the mechanism on meaning-centered coping, while practically, the findings provide valuable insights for educating training college students to leverage the wisdom of meaning theory to sustain their mental health in future challenges.

## 1. Introduction

In daily life, people often face various stressful situations that can impact their mental health. Particularly, when individuals face prolonged exposure to unremitting chronic stressors (e.g., pandemic-related uncertainties/quarantine, chronic health conditions, academic pressures, or financial strain) in the absence of effective coping strategies may lead to long-term physical and psychological exhaustion ([65]; [5]). This includes neuroendocrine dysregulation (such as the abnormal activation of the nervous system, impaired cognitive function, and rigid behavioral patterns) and depression, anxiety, loneliness, and post-traumatic stress ([36]; [2]). Emerging as an acute public health crisis, the COVID-19 pandemic has transitioned into a chronic stressor. Numerous studies have demonstrated its significant adverse effects on mental health, with increasing levels of anxiety, depression and post-traumatic stress symptoms (PTSSs) ([5]; [35]). In particular, prevalence rates of mental health disorders—particularly anxiety—are significantly higher among children and adolescents, with a substantial proportion exhibiting persistent and chronic symptom trajectories ([60]; [38]). Moreover, the meaning-making theory suggests that chronic stress continues to disrupt normative life functioning and occupational performance, eroding established systems of meaning-making and goal-directed behaviors that underpin psychological resilience. Especially when people collectively experience long-term chronic stress, such as during an epidemic outbreak, collective coping often becomes necessary. Therefore, it is particularly important to explore how college students in a collectivist cultural context can proactively cope with stress, effectively utilize self-awareness and environmental resources to buffer against pressure, and reduce anxiety and depression caused by stress. This study aimed to examine the impact of meaning-centered coping strategies on the negative psychological health indicators (primarily anxiety and depression) related to chronic stress among college students. It also explored whether variables associated with collectivist culture and collective coping (such as interdependent self-construal and school connectedness) play a significant mediating role, aiming to uncover the underlying mechanisms.

### 1.1. Meaning-Centered Coping and Mental Health of College Students

The Stress and Coping Theory posits that when individuals encounter stressful situations, they experience a series of negative physical and psychological reactions. To alleviate these reactions, individuals engage in cognitive appraisal and adopt corresponding coping strategies. These coping strategies primarily include problem-focused coping and emotion-focused coping, which involve either resolving the stressor itself or regulating one’s emotions to relieve stress ([29]). However, these strategies generally offer only temporary relief and might even intensify negative emotions after the initial alleviation of stress ([24]; [9]). When individuals are exposed to chronic stressors, they are unable to completely eliminate the stressors, nor can they effectively regulate the negative emotions triggered by stress through emotion-focused coping. This situation not only leads to anxiety regarding an uncontrollable future, real-life survival challenges, or financial pressures, keeping individuals in a persistently negative psychological state ([18]; [28]), but also undermines their pursuit of long-term goals and achieving a sense of value to achieve life meaning, ultimately triggering an existential crisis ([51]).

Coping with chronic, uncontrollable, or progressively worsening stressful situations, Folkman introduced the concept of “meaning-focused coping”. This strategy involves evaluating the situation through one’s beliefs (e.g., religion, beliefs), values, and life goals to maintain well-being and find potential benefits and meanings in stressful experiences ([42]). It is a proactive and creative approach to stress, viewing difficulties as challenges and transforming adversity into opportunities for personal growth ([62], [63]; [71]). This coping mechanism is supported by research in positive psychology, which shows that positive emotions can arise even in highly stressful or traumatic situations ([1]). Having positive emotions can help individuals think and perceive situations in a more optimistic way ([11]). When faced with challenges, those who adopt meaning-focused coping will more use the strategies of cognitive reappraisal, finding benefits, repeating benefits, adjusting goals, reordering priorities, and injecting positive meanings into daily events, which help to establish and adaptability ([19]; [12]). Based on the perspectives of the sense of meaning and coping, two theoretical frameworks have progressively emerged: the meaning-making model and the meaning maintenance model ([41]; [17]). Both theories propose that individuals are constantly engaged in actively constructing or seeking meaning within contexts. Meaning can be achieved by actively aligning one’s global meaning with situational meaning through assimilation or accommodation, or it can be reconstructed via compensatory mechanisms in other domains when one’s existing sense of meaning is disrupted. More recently, [10] ([10]) integrated the principles of the meaning-making theory and Frankl’s views on creating meaning to transcend collective trauma, proposing the strategy of meaning-centered coping. They define meaning-centered coping as a set of emotional, cognitive, and behavioral strategies. It involves accepting harsh realities, affirming the meaning of life, incorporating others into coping motives and ways, and constructing coping mechanisms of positive personal and world meaning from immutable circumstances. These coping mechanisms include positive reconstruction, hope, existential courage, an appreciation of life, engagement in meaningful activities, and prosocial behavior. Against the backdrop of the global experience of the pandemic, studies in 30 countries around the world consistently demonstrated that meaning-centered coping was the most robust positive predictor of physical health and well-being during the pandemic. It is also the strongest negative predictor of psychological distress (depression, anxiety, stress, and loneliness), significantly more effective than problem- and emotion-focused coping. Research on meaning-making intervention also indicates that meaning reconstruction can help patients with chronic illnesses overcome disease-related anxiety ([20]; [57]). Merely having a sense of meaning in life can serve as a protective factor for mental health. The sense of meaning in life has buffered the specific conditions of stress, boredom, depression, and anxiety, which are caused by the pandemic ([55]; [60]).

### 1.2. School Connectedness as a Mediator

Schools are an important living space for college students, and the school environment is inevitably a crucial source for them to obtain coping resources. In a collectivist cultural context, when faced with collective and chronic stress situations, schools tend to adopt more enduring and widespread collective coping strategies. Variables related to the school, particularly students’ emotional connection to the school, may influence their mental health status. School connectedness is defined as the students’ perception of acceptance, inclusiveness, respect, and support in their school. It encompasses their positive attitudes toward school, a sense of belonging, and emotional connections with both peers and teachers ([15]; [70]). Previous studies show that strong school connectedness enhances students’ cognitive, emotional, and behavioral development ([66]). Furthermore, students who feel a greater connection to their school are more likely to follow rules, achieve higher grades, experience greater life satisfaction and well-being, enjoy better mental health, exhibit more positive psychological traits, and engage in fewer risky behaviors ([39]). Research during the pandemic revealed a strong negative relationship between school connectedness and symptoms of anxiety and depression, persisting even after 13 weeks of online learning ([43]). This indicates that school connectedness plays a crucial role in protecting students’ mental health.

College students’ proactive use of meaning-centered coping strategies influences their school connectedness. Meaning-centered coping includes strategies such as actively involving others in coping, participating in meaningful activities, and engaging in prosocial behaviors, all of which can enhance school connectedness. Individuals who utilize these coping strategies more frequently are more likely to develop a stronger sense of school connectedness. Research indicates that individuals who actively seek and find meaning in life are more likely to form positive relationships and feel a greater sense of connectedness ([50]; [21]; [52]). Students with a strong sense of meaning also perceive a greater connection to parents, peers, teachers, and schools, which may encourage them to become more connected with their school in their actions and decisions ([68]). Overall, school connectedness may act as a mediator in the relationship between meaning-centered coping and mental health.

### 1.3. Interdependent Self-Construal as a Mediator

Meaning-centered coping refers to the process of constructing positive personal and global meanings from immutable situations. According to the meaning-making theory, individuals compare, evaluate, and assess the sense of meaning derived from a situation against their pre-existing belief systems and value hierarchies ([41]), thereby reconstructing new meanings through continuous adjustment and revision. It is evident that meaning reconstruction fundamentally relies on individuals’ cognition of themselves and their circumstances. Individuals employing meaning-centered coping typically need external resources (such as collective or interpersonal support) to navigate adversity. This process often involves participating in collective or prosocial activities, establishing interpersonal relationships, and seeking social support ([41]; [10]). Consequently, individuals’ interdependent self-construal (self-cognition relating to others or collectives) may serve as a mediating variable through which meaning-centered coping influences school connectedness and mental health.

Individuals with interdependent self-construal tend to define themselves through their relationships with others (e.g., intimate partners) or collective affiliations (e.g., professional groups) ([33]), deriving a stable sense of self-worth by cultivating harmonious interpersonal relationships ([7]). Interdependent self-construal is an important self-cognitive factor in stress coping and situation adaptation. Research has shown that interdependent self-construal is significantly associated with stress response in both daily hassles and chronic stress, critically influencing mental health ([7]). In daily stress situations, individuals with interdependent self-construal prioritize relational dynamics and are able to obtain emotional and material support from their social networks ([54]). This process buffers the negative effects of stress on mental health. In chronic stress situations, interdependent self-construal helps individuals maintain psychological balance and well-being by adhering to social norms and strengthening their dependence on others ([23]; [56]; [25]). In collectivist culture contexts, the higher an individual’s level of interdependent self-construal, the higher their level of mental health ([22]; [27]).

An individual’s use of meaning-centered coping may strengthen their interdependent self-construal, especially in chronic stress situations. First, the persistence of chronic stress makes it difficult to cope with individually, often necessitating collective coping or reliance on support from others. This is particularly true when people experience stress collectively, as these stressful situations are typically relational, necessitating a collective response that transcends individual efforts (e.g., collective prevention measures during a pandemic, social assistance during economic hardships) ([46]). When individuals engage in collective coping or receive social support from others, they reinterpret their relationships with the collective, others, and the world, reconstructing a sense of meaning. This process reinforces their interdependent self-construal. In the collective response to chronic stress, collectivist cultures place even greater emphasis than usual on the values and behavioral norms of mutual interdependence. According to the dynamic theory of interdependent self-construal, interdependent self-construal evolves dynamically with cultural contexts, situational demands, and individual social cognition ([58]; [71]). Second, positive emotions (e.g., optimism, hope) generated by meaning-centered coping strengthen trust in interpersonal relationships among those with interdependent self-construal. This, in turn, promotes their emphasis on the values of collective coping or relationships, further enhancing their interdependent self-construal. However, research on the relationship between meaning-centered coping and interdependent self-construal remains limited.

### 1.4. The Chain Mediating Effect

The interdependent self-construal of college students is closely related to their school connectedness. For college students experiencing chronic stress, the school serves as a primary living environment and a vital source of coping resources. Research indicates that individuals with higher interdependent self-construal tend to engage in holistic perceptual processing, more readily noticing contextual, environmental, and social information and forming complex cognitive representations ([71]). These processes serve as the foundation for students to focus on campus environmental factors, integrate their perceptions of relationships or emotions with teachers and peers, and ultimately foster a sense of school belonging. Additionally, the needs of individuals with interdependent self-construal are often related to others and society, driving them to actively establish and maintain interpersonal bonds, gaining familial affection, romantic love, or a sense of collective belonging and positive emotions such as trust, support, and connection ([37]; [22]). It is evident that college students with higher levels of interdependent self-construal are more likely to perceive or establish connections with their school. In other words, an individual’s interdependent self-construal influences their perception of school connectedness. Furthermore, meaning-centered coping is primarily achieved through core cognitive processes of self and situational evaluation and cognitive restructuring, which engage or activate the cognitive schemas associated with interdependent self-construal. This cognitive process facilitates an individual’s perception of emotional connections with the school, teachers, classmates, and others. This mechanism reflects the fundamental psychological principle of the progression from cognition to emotion. Therefore, interdependent self-construal and school connectedness play a chain mediating role in the process by which meaning-centered coping impacts mental health.

### 1.5. The Current Research

Currently, researchers have separately examined the effects of meaning-centered coping, interdependent self-construal, and school connectedness on chronic stress-related depression, anxiety, and behavioral outcomes ([10]; [13]; [43]). Few studies have simultaneously investigated the relationships among these four variables or constructed a theoretical model integrating them. Based on practical observations during the pandemic in a collectivist culture and the theoretical reasoning outlined above, the current study presents four research hypotheses. Hypothesis 1 suggests that meaning-centered coping negatively affects the depression and anxiety of college students. Hypothesis 2 indicates that school connectedness mediates the relationship between meaning-centered coping and mental health. Hypothesis 3 proposes that students’ interdependent self-construal also acts as a mediator between meaning-centered coping and mental health. Finally, Hypothesis 4 posits that in the process by which meaning-centered coping affects mental health, college students’ interdependent self-construal plays a chain mediating role by affecting school connectedness.

## 2. Materials and Methods

### 2.1. Participants

This study involved undergraduate students from a university in Qinghai Province who stayed on campus for approximately three months during the epidemic. A stratified random sampling method was used to select 1028 students from freshmen to juniors across various majors including arts, agriculture, engineering, and medicine, after obtaining their informed consent. The recruitment of participants and the completion of the questionnaire lasted for one week. The survey was conducted using the Questionnaire Star platform, and the estimated time to complete all the questionnaires was approximately 30 min. After excluding responses from participants who provided consistent answers or for whom their answering time was too short, 856 responses were deemed valid, resulting in a 93% effectiveness rate. The demographic breakdown consisted of 329 freshmen, 278 sophomores, 249 juniors, 328 males, and 528 females. Ethnically, 492 participants were Han, 177 Tibetan, 102 Hui, and 85 from other ethnic backgrounds; 126 were studying liberal arts, 164 medicine, 411 engineering, and 155 agriculture. There were 359 students from urban areas and 497 from rural areas; only 164 were children, and 692 were not. This study collected data anonymously and obtained informed consent in advance, in compliance with relevant ethical requirements.

### 2.2. Measurements

Meaning-centered coping: The Meaning-Centered Coping Scale (MCCS), developed by [10] ([10]), comprises nine items that measure various coping strategies such as positive restructuring, maintaining life appreciation and hope, adopting a courageous attitude against adversity, and engaging in prosocial and meaningful activities. Examples of these items include statements like “I have found a personal meaning in the current situation”. Participants evaluated these statements on a 7-point Likert scale, ranging from 1 (I do not agree at all) to 7 (I completely agree). The MCCS has been validated across 30 countries, which demonstrated a robust single-factor structure and exhibited high test–retest reliability. The current study also demonstrated a robust single-factor structure (χ^2^/df = 150.85/28, RMSEA = 0.068, CFI = 0.91,TLI = 0.88), and Cronbach’s α of this study is 0.90.

Interdependent self-construal: The interdependent self-construal subscale from the Self-Construal Scale was employed in this study ([49]; [59]). Responses are measured on a 7-point Likert scale, ranging from 1 (strongly disagree) to 7 (strongly agree): the higher the total score, the more inclined one is toward interdependent self-construal. Research indicates that the scale has shown robust reliability and validity among Chinese participants; the retest reliability after one month was 0.79 ([59]; [26]). For this research, Cronbach’s α of the interdependent subscale is 0.87.

School connectedness: The School Connectedness Scale was employed in this study ([67]). It includes 10 items across three dimensions: peer support, teacher support, and a sense of belonging to the school. Example items are “My classmates can really help me” for peer support, “I feel my teachers care about me” for teacher support and “I feel a part of the school” for school belonging. The scale uses a five-point rating system ranging from 1 (strongly disagree) to 5 (strongly agree), with higher scores reflecting a stronger sense of school connectedness. Studies have demonstrated the scale’s reliability and validity among Chinese college students (the main fit indices of the EFA: CFI = 0.98, TLI = 0.97, RMSEA = 0.08) ([14]; [66]). In the current study, Cronbach’s α for the scale was 0.77.

Depression: The participants’ depression levels were measured using the revised Simplified Center Epidemiological Studies of Depression Scale (SCESDS) ([69]). This scale includes 13 items and consists of 3 dimensions: physical symptoms (5 items, such as “I have trouble sleeping”), depressive mood (5 items, such as “I feel downhearted”), and positive emotions (3 items, such as “I feel happy with my life”). Each item was rated on a 4-point scale from 0 (less than one day) to 3 (more than five days), reflecting how frequently the participants experienced each symptom over the past week. A higher total score indicates more pronounced depressive symptoms. Previous research has demonstrated that this questionnaire possesses good reliability and validity when used among Chinese participants (the main fit indices from CFA were CFI = 0.98, NFI = 0.98, NNFI = 0.97, GFI = 0.97, and RMSEA = 0.058) ([69]). In this study, the scale demonstrated good reliability with a Cronbach’s α of 0.82.

Anxiety: The Beck Anxiety Inventory (BAI) was using in this study ([4]; [40]). It consists of 21 self-assessment items, and each item represents an anxiety symptom (like tremors in the legs and dizziness) and is scored on a 4-point scale: 0 (none), 1 (mild, not much bother), 2 (moderate, discomfort is tolerable), and 3 (severe, barely tolerable). Participants were asked to rate the frequency of each symptom over the past week. The higher the score, the more pronounced the symptoms of anxiety. This scale has demonstrated good reliability and validity among Chinese participants ([40]). In the current study, Cronbach’s α was 0.96.

### 2.3. Statistical Analyses

The data were analyzed using SPSS 27.0 software. Initially, Mahalanobis distance was employed to identify outliers, and Harman’s single-factor method was applied to assess common method bias. Subsequently, descriptive statistics were conducted. Furthermore, the SPSS macro plug-in PROCESS MODEL 6, developed by [16] ([16]), was utilized to test the chain mediation effect.

Statistical power is evaluated using the Monte Carlo simulation method ([45]). The results from 10,000 repeated experiments indicate that, to achieve a statistical power level of 0.8, all path coefficient estimates reach the desired power level of 0.95 with a sample size of at least 453 participants.

## 3. Results

### 3.1. Descriptive Statistics and Correlation Analysis

The findings from the descriptive statistical analysis and correlation analysis are summarized in Table 1. These results show that meaning-centered coping, interdependent self-construal, and school connectedness are significantly and negatively correlated with mental health (anxiety and depression) (*r* = −0.18~−0.31, *p* < 0.01). Additionally, there is a strong positive correlation between meaning-centered coping, interdependent self-construal, and school connectedness (*r* = 0.56~0.65, *p* < 0.01).

### 3.2. Test of Chain Mediation Effect

Variables were standardized. The variance inflation factors (VIFs) for all predictor variables in this study are below 2.06, indicating no multicollinearity issues ([48]). The regression results indicate that meaning-centered coping is a negative predictor of depression and anxiety (β = −0.25, *t* = −7.53, *p* < 0.001; β = −0.26, *t* = −7.76, *p* < 0.001). Combined with the results of the correlation analysis, this suggests that it is suitable for conducting a chain mediation effect analysis. After controlling for gender, grade, and ethnicity, the chain mediation effect was then analyzed using Model 6 from the PROCESS plug-in for SPSS. The 95% confidence intervals for the mediation effects were estimated using 5000 Bootstrap samples. As shown in Figure 1, meaning-centered coping emerged as a significant positive predictor of interdependent self-construal (β = 0.64, *t* = 24.53, *p* < 0.001) and school connectedness (β = 0.27, *t* = 7.97, *p* < 0.001) and a negative predictor of depression and anxiety (β = −0.16, *t* = −3.71, *p* < 0.001; β = −0.12, *t* = −2.72, *p* < 0.01), which supports hypothesis 1. Additionally, interdependent self-construal positively predicted school connectedness (β = 0.44, *t* =12.68, *p* < 0.01) and depression (β = 0.10, *t* = 2.23, *p* < 0.05) but did not significantly affect anxiety (β = −0.02, *t* = −0.46, *p* > 0.05), and school connectedness was a significantly negative predictor of depression and anxiety (β = −0.28, *t* = −6.56, *p* < 0.01; β = −0.22, *t* = −5.27, *p* < 0.01). The chain mediation models explains 11% of the variance in depression and 12% of the variance in anxiety.

The analysis of mediating effects is presented in Table 2. The results indicate that the impact of meaning-centered coping on depression is significant through two pathways. Specifically, the Bootstrap 95% confidence intervals for the mediating effects of interdependent self-construal and school connectedness have two paths that do not contain 0. The total mediating effect value is −0.09, accounting for 36% of the total effect (−0.25). (1) Indirect effect 1: meaning-centered coping → school connectedness → depression, accounting for 32% of the total effect, and (2) indirect effect 2: meaning-centered coping → interdependent self-construal → school connectedness → depression, with the mediating effect accounting for 32% of the total effect. The impact of meaning-centered coping on anxiety is also significant through two pathways, mediated by interdependent self-construal and school connectedness. The Bootstrap 95% confidence intervals for these two paths do not contain 0, and the mediating effect value is −0.14, accounting for 54% of the total effect (−0.26). (1) Indirect effect 1: meaning-centered coping → school connectedness → anxiety, accounting for 23% of the total effect, and (2) indirect effect 2: meaning-centered coping → interdependent self-construal → school connectedness → anxiety, with the mediating effect also accounting for 23% of the total effect. However, the 95% confidence intervals for the independent mediating effect of interdependent self-construal contain 0, indicating that this particular mediation effect is not statistically significant. Consequently, the results support hypotheses 2 and 4 but do not support hypothesis 3.

## 4. Discussion

### 4.1. The Relationship Between Meaning-Centered Coping and the Mental Health of College Students

The results of this study indicate that meaning-centered coping strategies negatively predict depression and anxiety among college students, corroborating previous research ([47]; [10]; [3]). Such coping strategies reliably mitigate anxiety and depression in the context of chronic stress. When college students use meaning-centered coping, they not only experience positive emotions like hope and optimism by reconstructing or redefining meaning to alleviate stress but also gain enduring motivation and more readily reallocate various coping resources. This is its advantage over problem- and emotion-focused coping strategies ([11]; [60]). Meaning reconstruction enables individuals to generate new ideas about self-perception, interpersonal relationships, and life philosophies, offering intrinsic advantages such as positive evaluation, benefit discovery, and reflective and dialectical thinking. These advantages assist individuals in growing through adversity, enhancing their psychological resilience to alleviate depression and anxiety more sustainably ([32]; [60]).

### 4.2. The Mediating Role of Interdependent Self-Construal and School Connectedness

This study found that school connectedness among college students plays a partial mediating role between meaning-centered coping and mental health (depression and anxiety). This suggests that meaning-centered coping not only directly exerts a negative effect on students’ mental health but also indirectly affects it through their school connectedness. According to the developmental contextual theory, individual development is influenced by both family and school environments ([30]). During the chronic stress of the pandemic, college students were primarily engaged in on-campus activities, making the school environment and its related resources the most direct and accessible coping resources. Moreover, under a collectivist culture, schools also adopt sustained collective coping strategies. When college students utilize meaning-focused coping, they may find it easier to perceive their connection with the school as a collective entity and reassess and perceive their relationship with the school and its faculty and students during special periods, thereby influencing their sense of connection with the school. Furthermore, research indicates that an individual’s sense of meaning can enhance their sense of social connectedness: those who actively seek meaning in life are more inclined to view social connections as important goals; they are also more motivated to engage in social connections and related activities; and they exhibit greater social appeal and are more welcomed by others, which in turn increases their social connectedness and integration ([53]; [50]). In conclusion, college students who cultivate a sense of meaning through meaning-centered coping experience enhanced school connectedness. This strong sense of connectedness acts as a protective factor, reducing the risk of depression and anxiety and supporting students’ mental health. Specifically, this pertains to when students actively participate in campus life, demonstrate prosocial behaviors, and make proactive efforts to strengthen their relationships with teachers and peers. They foster positive social relationships within the school community. This process cultivates a strong sense of belonging, support, and security. Such connectedness improves their well-being and reduces stress-related depression and anxiety (such as separation from family and friends, monotonous campus routines, and uncertainty about the future) ([31]; [6]).

This study also revealed a strong correlation between meaning-centered coping and interdependent self-construal, with the former positively predicting the latter. However, mediation effect analysis indicates that interdependent self-construal does not independently mediate the relationship between meaning-centered coping and depression or anxiety among college students, thus failing to support Hypothesis 3. Firstly, this result demonstrates the close association between meaning-centered coping and interdependent self-construal, aligning with previous research findings on the sense of life meaning and interdependent self-construal ([8]). This research found that individuals with interdependent self-construal perceive the self as a malleable and fluid entity, which is associated with the continuous search for meaning. Meaning-centered coping involves strategies related to others or collectives, such as seeking support from others, incorporating others into one’s coping motivations. Furthermore, during prolonged stressors such as an epidemic, individuals must consistently adhere to the collective coping measures mandated by public health authorities. When adopting coping strategies related to others or conforming to collective coping, according to the meaning-making theory, individuals continuously evaluate, balance, and adjust between situational meanings and their pre-existing personal meanings ([72]). This process reconstructs a sense of meaning connected to others, collective goals, and values (e.g., human interdependence). It may be intrinsically linked to interdependent self-construal (individuals fundamentally conceptualize the self as embedded within relational and collective frameworks). In addition, some researchers believe that meaning-focused coping is a cultural process, facilitated and constrained by cultural resources ([46]). According to this perspective, during periods of collective experience of disease outbreaks and chronic stress, collectivist cultures tend to emphasize and highlight the interdependent relationships between individuals and others or the collective. This can encourage individuals to reconstruct a sense of meaning related to the collective and interpersonal relationships. This process shares commonalities with the dynamic changes involved in interdependent self-construal. In the context of collectivist culture, the sociocultural norms promoted during collective coping provide individuals with cultural practices and behavioral routines integrated into their daily lives ([34]). Subsequently, these, in turn, foster automatic psychological responses of interdependence with others, thereby activating and shaping the individual’s interdependent self-construal. Secondly, interdependent self-construal did not play a mediating role in the relationship between meaning-centered coping and depression/anxiety, which could be attributed to its cognitive characteristics. Although previous studies have found that individuals with interdependent self-construal are more likely to perceive relational harmony, social support, self-esteem, and positive emotions, alleviating depression, anxiety, etc., ([22]; [64]), other studies also indicate that individuals with higher levels of interdependent self-construal place greater emphasis on context-related information, are more sensitive to socially evaluative threats within given contexts, experience higher stress levels in stressful situations, and are prone to suppressing emotional expression. Consequently, this lead to the higher levels of stress-related depression and anxiety ([44]). This study found a low negative correlation between interdependent self-construal and depression/anxiety. Future research could further explore the underlying mechanisms involved.

### 4.3. The Chain Mediation Effect of Interdependent Self-Construal and School Connectedness

The results of this study reveal that interdependent self-construal and school connectedness play a chain mediation role in the relationship between meaning-centered coping and the mental health (depression and anxiety) of college students, forming the following pathway: meaning-centered coping → interdependent self-construal → school connectedness → mental health (depression and anxiety). The results indicate a close relationship between college students’ interdependent self-construal and their school connectedness. Meaning-centered coping may reduce anxiety and depression through the holistic processing of the self and environment and cognitive reappraisal. As previously mentioned, when reconstructing a sense of meaning in stressful situations, it activates college students’ reassessment and reconstruction of their relationship with the self and the environment, which is characterized by interdependent self-construal. They perceive themselves as part of a social group or dependent on others, focusing on their social relationships, striving to maintain connections with others, and participating in collective or social activities ([34]). Based on this holistic self-construction, college students are more likely to notice relational cues from their school environment and are more sensitive to their sense of connection and belonging with teachers and classmates, which in turn influences their mental health (depression and anxiety). Furthermore, the process by which interdependent self-construal affects school connectedness aligns with the stress buffering hypothesis, which suggests that in situations of loss or threat, individuals who are more connected to others feel they have more support resources available ([61]).

The results of this study suggest that in the face of persistent and challenging chronic stress, mental health education for college students should implement a dual-track approach. Firstly, it should cultivate students’ ability to cope by seeking meaning, inspiring existential wisdom and bolstering psychological resilience. Secondly, it should reinforce education in interpersonal communication and adaptability within the campus, facilitating the formation of meaningful group relationships and social identities. These approaches would enable students to derive enhanced emotional support from campus life, thereby alleviating stress-related psychological distress.

### 4.4. Implications

In the context of a collectivist culture and the chronic stress situation, the present study investigates the positive impact of meaning-centered coping on the mental health of college students, primarily in reducing stress-related anxiety and depression. It further examines the chain mediating roles of interdependent self-construal and school connectedness. Theoretically, this study focuses on the impact of school contextual factors and self-perception factors (related to social and cultural aspects) on the mental health of college students, thereby validating the developmental contextual theory. Additionally, it analyzes the relationship between meaning reconstruction in response to chronic stress and interdependent self-construal within a collectivist cultural context and establishes a theoretical model. This study elucidates the intrinsic connection between individuals’ meaning-making and self-construction and posits that values, beliefs and behavioral patterns within sociocultural contexts may serve as a common foundation for such constructions. It validates and extends both the theory of meaning-making and the cultural dynamic theory of interdependent self-construction. Practically, this study clarifies how college students actively construct meaning when faced with insurmountable challenges, exploring from the perspective of interpersonal relationships how they identify and utilize school resources to undertake proactive coping mechanisms. It provides a practical approach for fostering college students’ ability to respond positively, enhancing their coping skills and psychological resilience in the future.

### 4.5. Limitations and Future Research

The current research has several limitations that will be addressed in future studies. First, this study explores the internal mechanisms of meaning-centered coping among college students within a collectivist cultural context. Future research should consider implementing cross-cultural (collectivist/individualistic culture) research to explore the associations between meaning-centered coping and factors such as self-construction and school connectedness. Second, this study employs a cross-sectional research design, with all data derived from self-reported retrospective accounts. It lacks dynamic experimental research and longitudinal designs. Future research should implement methods like situational activation or longitudinal designs to both obtain objective data and establish causal relationships. Third, this study only considered two factors—interdependent self-construal and school connectedness—in coping with adversity. However, coping with chronic stress may be related to various factors such as family and individual personality. Moreover, even within collectivist cultures, the coexistence of interdependent and independent self-construal is becoming more common. Future research should include multiple variables to provide a more comprehensive and in-depth exploration of the internal mechanisms of meaning-centered coping.

## 5. Conclusions

This study found that meaning-centered coping can effectively alleviate college students’ depression and anxiety and protect their mental health. This coping strategy is particularly beneficial due to the beliefs and meanings tied to interdependence and solidarity. These beliefs enhance students’ interdependent self-construal and their holistic perception of themselves and their surroundings. Consequently, students who felt a stronger connection to their school experienced lower levels of anxiety and depression, thereby preserving their mental health.

## Figures and Tables

**Figure 1 behavsci-15-00955-f001:**
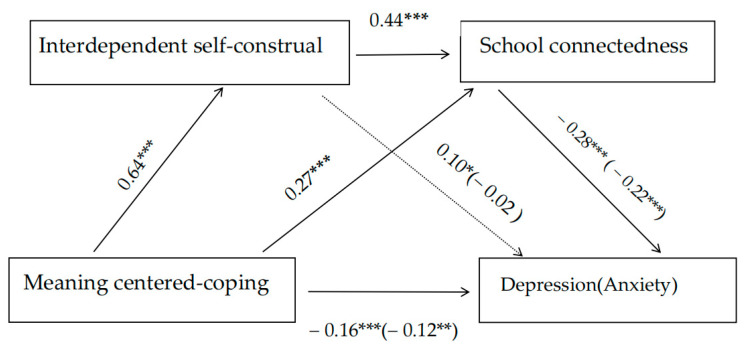
Chain mediation effect model. Note: *** *p* < 0.001; ** *p* < 0.01; * *p* < 0.05.

**Table 1 behavsci-15-00955-t001:** Descriptive statistics and correlation analysis.

Variable	M	SD	1	2	3	4	5	6	7	8
1. Meaning-centered coping	5.29	1.06	1							
2. Interdependent self-construal	5.29	0.83	0.65 **	1						
3. School connectedness	3.43	0.49	0.56 **	0.61 **	1					
4. Depression	0.97	0.50	−0.25 **	−0.18 **	−0.31 **	1				
5. Anxiety	0.41	0.52	−0.27 **	−0.25 **	−0.31 **	0.60 ***	1			
6. Gender	1.62	0.49	0.02	0.03	0.00	0.03	0.00	1		
7. Grade	1.91	0.82	−0.11 **	−0.15 **	−0.10 **	0.05	0.14 **	−0.04	1	
8. Ethnicity	1.80	1.04	−0.03	−0.01	0.04	−0.04	0.03	0.08 *	0.04	1

Note: *** *p* < 0.001, ** *p* < 0.01, * *p* < 0.05.

**Table 2 behavsci-15-00955-t002:** Test of mediating effect.

	Effect	Path	Coeff	Boot SE	Boot LLCI	Boot ULCI	Relative Mediating Effect of Total Effect
Depression	Indirect effect	Indirect effect 1	−0.08	0.02	−0.11	−0.04	32%
		Indirect effect 2	−0.08	0.02	−0.11	−0.05	32%
	Total indirect effect		−0.09	0.04	−0.12	−0.02	36%
	Direct effect		−0.16	0.04			
	Total effect		−0.25				
Anxiety	Indirect effect	Indirect effect 1	−0.06	0.02	−0.10	−0.03	23%
		Indirect effect 2	−0.06	0.01	−0.10	−0.04	23%
	Total indirect effect		−0.14	0.04	−0.21	−0.07	54%
	Direct effect		−0.12	0.04			
	Total effect		−0.26				

Note: Indirect effect 1: meaning-centered coping → school connectedness → depression/anxiety; indirect effect 2: meaning-centered coping → interdependent self-construal → school connectedness → depression/anxiety.

## Data Availability

The raw data supporting the conclusions of this article will be made available by the authors on request.

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
