# Peer review of "How Does Meaning-Centered Coping Influence College Students’ Mental Health? The Mediating Roles of Interdependent Self-Construal and School Connectedness"

_behavsci, 2025, doi:10.3390/bs15070955_

Round 1
Reviewer 1 Report
Comments and Suggestions for Authors
Title: How does meaning-centered coping influence college students’ mental health? The mediating roles of interdependent self-construal and school connectedness
Thank you for the opportunity to review this manuscript. Overall, the article addresses a relevant and timely topic, with significant implications for understanding coping and mental health among university students. It also provides valuable insights for the development of health promotion strategies targeted at this population.
Below are my observations, organized by section:
- General Assessment
The manuscript presents a rigorous methodological design, employs valid and reliable psychometric instruments, and offers a relevant theoretical contribution by integrating psychological and social variables within a chain mediation model.
- Introduction
- The authors clearly identify a gap in the literature regarding the joint analysis of meaning-centered coping, interdependent self-construal, and school connectedness.
- The theoretical background is thorough and up to date. However, the introduction lacks sufficient contextual information to better support the interpretation of results.
- The sociocultural context, especially the collectivist framework mentioned in the discussion, should be introduced and defined earlier in the manuscript.
- The section refers to mental health in a general way; a more in-depth treatment of this concept and its relevance in the university context is recommended.
- Method
- The instruments used are appropriate for the study objectives. However, detailed information regarding the validity of most scales (interdependent self-construal, school connectedness, depression, anxiety) is missing. This is essential to assess their relevance and robustness.
- A more comprehensive description of each instrument is needed, including their components or subscales, scoring methods, and interpretation guidelines.
- The procedure section should be expanded to explain how participants were recruited (beyond the sampling technique), the context in which they were surveyed, their sociocultural characteristics, the duration of the recruitment process, and whether any protocol was followed for identifying participants with potential mental health concerns.
- Ethical considerations are not reported and must be included.
- Results
- The statistical analyses are well presented and appropriate for the hypotheses tested.
- Discussion
- The discussion is coherent and grounded in the results.
- References are relevant, although the number of self-citations from the research team should be reviewed.
- A deeper discussion of the cultural conditions (e.g., collectivist orientation) would enhance the interpretation of results. This can be improved by incorporating more contextual information in the introduction and methods.
Specific Comments
- Lines 295–301: The chain mediation model is presented, but the current figure should be enhanced with greater detail for clarity.
- Table 2: Well organized, but a brief interpretation below the table is recommended.
- Figure 1: Clear, but too simple. It is suggested to add β coefficients and significance values within the diagram.
- Methods (lines 274–279): SPSS and PROCESS were used, but no mention is made of whether multicollinearity was tested. This should be clarified.
- Limitations (lines 452–467): Adequately discussed. Emphasizing the need for longitudinal designs to establish causality is recommended.
General Recommendation
Accept after minor revisions
Author Response
Comments 1: The authors clearly identify a gap in the literature regarding the joint analysis of meaning-centered coping, interdependent self-construal, and school connectedness. The theoretical background is thorough and up to date. However, the introduction lacks sufficient contextual information to better support the interpretation of results. The sociocultural context, especially the collectivist framework mentioned in the discussion, should be introduced and defined earlier in the manuscript. The section refers to mental health in a general way; a more in-depth treatment of this concept and its relevance in the university context is recommended.
Response1: (â… ) Based on the review comments, we have made revisions to both the introduction and discussion sections, emphasizing the significance of this study in the context of collectivism and highlighting the intrinsic connections established between the variables. (â…¡) Based on the reviewer’s comments, the introduction and discussion sections of this paper have clarified the defined mental health indicators, with a particular emphasis on alleviating negative indicators during the discussion. Additionally, inappropriate references to “mental health” have been revised. However, many of the cited existing studies also directly adopt the broad concept of “mental health”, which may still include some unclear definitions. We kindly ask for the reviewer’s understanding regarding this matter.
Comments 2: (â… ) The instruments used are appropriate for the study objectives. However, detailed information regarding the validity of most scales (interdependent self-construal, school connectedness, depression, anxiety) is missing. This is essential to assess their relevance and robustness. A more comprehensive description of each instrument is needed, including their components or subscales, scoring methods, and interpretation guidelines. (â…¡) The procedure section should be expanded to explain how participants were recruited (beyond the sampling technique), the context in which they were surveyed, their sociocultural characteristics, the duration of the recruitment process, and whether any protocol was followed for identifying participants with potential mental health concerns. Ethical considerations are not reported and must be included.
Response2: (â… ) Among the various scales used in the research, only the Meaning-Centered Coping Scale had not previously undergone an evaluation of its reliability and validity for use with Chinese participants. Therefore, this study conducted a confirmatory factor analysis to test its validity. The other scales used have been widely applied in studies with Chinese participants, and their reliability and validity have already been verified, so only their reliability was reported here. Based on the reviewers’ comments, we have supplemented the composition and sample items of each research scale, along with scoring methods and interpretation guidelines. Additionally, we consulted literature to illustrate the validity of different research scales as verified in previous studies. (â…¡) Based on the reviewer’s feedback, additional details regarding participants’ demographic variables and the recruitment process have been included. The study primarily aims to describe the relationship between participants’ coping mechanisms and mental health variables, without identifying any potential psychological issues of the participants. Before this study began, the school’s mental health institutions had already conducted assessments and counseling within the context of the pandemic. Therefore, participants with severe psychological issues were generally excluded in advance. The ethical review status of the study has been declared in accordance with the journal’s submission requirements and has also been briefly described in the Methods section.
Comments 3: (â… ) References are relevant, although the number of self-citations from the research team should be reviewed. (â…¡) A deeper discussion of the cultural conditions (e.g., collectivist orientation) would enhance the interpretation of results. This can be improved by incorporating more contextual information in the introduction and methods.
Response3: (â… ) In accordance with the review comments, the self-citation frequency of our research team’s literature has been examined. This topic represents the innovative research of our team within the context of the pandemic, and the self-citation count is zero. (â…¡) When discussing the impact of meaning-centered coping on interdependent self-construal and school connectedness, we have strengthened the discussion within the context of collectivist culture and the collective response to chronic stress.In terms of research implications, we have also summarized the theoretical significance of establishing the intrinsic relationships among the variables of this study within the context of collectivist cultures.The methods section did not address additional cultural considerations, as the cultural differences among the college student participants in this study were minimal.
Comments 4: Lines 295–301: The chain mediation model is presented, but the current figure should be enhanced with greater detail for clarity.
Response 4: According to the review comments, We have redrawn the figure to improve its clarity.
Comments 5: Table 2: Well organized, but a brief interpretation below the table is recommended.
Response 5: Based on the reviewers’ comments, we have marked the paths of the mediating effects below the table.
Comments 6: Figure 1: Clear, but too simple. It is suggested to add β coefficients and significance values within the diagram.
Response 6: To present the results of the mediation effect more clearly, We have added Table 2, which presents the analysis of the chain mediation model. It contains the regression β coefficients and significance values between variables.
Comments 7: Methods (lines 274–279): SPSS and PROCESS were used, but no mention is made of whether multicollinearity was tested. This should be clarified.
Response 7:Based on the review comments, we supplemented the multicollinearity test. The results indicate that multicollinearity does not exist, and these findings have been presented in the text.
Comments 8: Limitations (lines 452–467): Adequately discussed. Emphasizing the need for longitudinal designs to establish causality is recommended.
Response 8: We have supplemented the discussion on the limitations of cross-sectional studies in establishing causal inferences in the research limitations section of this paper.
Reviewer 2 Report
Comments and Suggestions for Authors
23.05.2025
How does meaning-centered coping influence college students’ mental health? The mediating roles of interdependent self-construal and school connectedness
The paper addresses an issue of significant scientific and practical importance and offers a well-organized and extensively researched contribution to the area. The purpose of this review is to provide helpful criticism that could improve the manuscript's coherence, clarity, and scientific rigor.
- Meaning-making theory encompasses several models and perspectives, including positive psychology, and stress-coping theories. In their paper, the authors cited the Meaning-focused Coping construct and Stress and Coping Theory. The authors should, in my opinion, expand on this framework, list pertinent authors in the field, and examine additional pertinent empirical research that is in line with this viewpoint and centered on clinical and health psychology.
- Since young adults who are college students make up the sample, information about the sociodemographics and developmental characteristics of this group should be included and discussed in addition to the study's findings. Are there any characteristics of this group that might be offered as possible justifications for the outcomes (particularly those that the Hypotheses did not preview)?
- The authors present a chain mediating model with two sequential mediators tested in a cross-sectional self-report survey study. These studies have some significant limitations even though they offer a valuable starting point for comprehending a construct or phenomenon. Although cross-sectional study design is not a constraint in and of itself, my choice was influenced by this as well as mediation testing. Please be aware that the inability to test temporal relationships between variables limits the use of mediation analysis on cross-sectional data. These weaknesses should be reported in the Limitations section.
- In several sections of the text, there are no citations. For instance, the text between lines 194–204 and 209–212. Please review the entire document and, if necessary, add the proper authors.
- Why did the author only perform a CFA to Meaning-Centered Coping Scale? Did the other measures have been already adapted/validated in previous studies with similar populations?
- The authors brought to discussion some theories (i.e., developmental contextual theory, dynamic theory of interdependent self-construal). This theoretical contribution is very important to the field, but in my opinion the theories should be more “linked” to the results and with the perspectives already presented in the Introduction. What do these theories add to the previous ones and what do the results contribute – or not - to the reinforcement of the theories?
Author Response
Comments 1: Meaning-making theory encompasses several models and perspectives, including positive psychology, and stress-coping theories. In their paper, the authors cited the Meaning-focused Coping construct and Stress and Coping Theory. The authors should, in my opinion, expand on this framework, list pertinent authors in the field, and examine additional pertinent empirical research that is in line with this viewpoint and centered on clinical and health psychology.
Response 1: Based on the reviewers’ suggestions, we provided a concise overview of Meaning-making theory and its related theories while listing the relevant authors when discussing meaning-centered coping. Additionally, we incorporated research on the relationship between meaning-centered coping and culture for further elaboration. However, there is currently a limited number of empirical studies in clinical and health psychology specifically centered on meaning-centered coping. Most studies on the relationship between a sense of meaning in life and mental health cannot be directly cited.Therefore, we did not include additional empirical research for support at this time. Going forward, we will continue to monitor research in this field and aim to conduct more in-depth and detailed studies in the future.
Comments 2: Since young adults who are college students make up the sample, information about the sociodemographics and developmental characteristics of this group should be included and discussed in addition to the study's findings. Are there any characteristics of this group that might be offered as possible justifications for the outcomes (particularly those that the Hypotheses did not preview)?
Response 2: Based on the reviewers’ comments, we have supplemented additional demographic information about the participants in this study. However, the demographic differences among participants do not explain the study’ findings. In the future, we can enhance the interpretation of the research findings by incorporating more information about participants’ family backgrounds or expanding the study to include participants from different regions.
Comments 3: The authors present a chain mediating model with two sequential mediators tested in a cross-sectional self-report survey study. These studies have some significant limitations even though they offer a valuable starting point for comprehending a construct or phenomenon. Although cross-sectional study design is not a constraint in and of itself, my choice was influenced by this as well as mediation testing. Please be aware that the inability to test temporal relationships between variables limits the use of mediation analysis on cross-sectional data. These weaknesses should be reported in the Limitations section.
Response 3:The limitations of cross-sectional studies in establishing causal inferences have been further discussed in the limitations section of this paper.
Comments 4: In several sections of the text, there are no citations. For instance, the text between lines 194–204 and 209–212. Please review the entire document and, if necessary, add the proper authors.
Response 4: Based on the review comments, the entire document's references have been re-examined, and gaps have been identified and addressed.
Comments 5: Why did the author only perform a CFA to Meaning-Centered Coping Scale? Did the other measures have been already adapted/validated in previous studies with similar populations?
Response 5: Among the various scales used in the research, only the Meaning-Centered Coping Scale had not previously undergone an evaluation of its reliability and validity for use with Chinese participants. Therefore, this study conducted a confirmatory factor analysis to test its validity. The other scales used have been widely applied in studies with Chinese participants, and their reliability and validity have already been verified, so only their reliability was reported here. Based on the reviewers’ comments, we have supplemented the composition and sample items of each research scale, along with scoring methods and interpretation guidelines. Additionally, we consulted literature to illustrate the validity of different research scales as verified in previous studies.
Comments 6: The authors brought to discussion some theories (i.e., developmental contextual theory, dynamic theory of interdependent self-construal). This theoretical contribution is very important to the field, but in my opinion the theories should be more “linked” to the results and with the perspectives already presented in the Introduction. What do these theories add to the previous ones and what do the results contribute – or not - to the reinforcement of the theories?
Response 6: In the implications section of this study, we summarize the contributions of our findings to previously introduced theories, such as developmental contextual theory and meaning-making theory.
Reviewer 3 Report
Comments and Suggestions for Authors
Thank you for this article. Its structure meets all the requirements for such a work, and its execution is also appropriate.
However, the ethics of the research must be clearly stated in the article itself (in its methodological part). It is likely that this was followed in the research, as it is mentioned in the paragraph below the article, i.e. Institutional Review Board Statement. This point should only contain the information requested by the Institutional Review Board Statement.
It is also advisable to remove the appendices containing the questionnaires. The authors did not create them themselves. The originals of these instruments do not appear as presented, which raises reasonable doubts about the presentation of such instruments.
Author Response
Comments 1: Thank you for this article. Its structure meets all the requirements for such a work, and its execution is also appropriate. However, the ethics of the research must be clearly stated in the article itself (in its methodological part). It is likely that this was followed in the research, as it is mentioned in the paragraph below the article, i.e. Institutional Review Board Statement. This point should only contain the information requested by the Institutional Review Board Statement.
Response 1: Thank you for your affirmation of this article and your suggestions regarding ethics. In combination with the editor’s comments, the ethical review status of the study has been declared in accordance with the journal’s submission requirements and has also been briefly described in the Methods section.
Comments 2: It is also advisable to remove the appendices containing the questionnaires. The authors did not create them themselves. The originals of these instruments do not appear as presented, which raises reasonable doubts about the presentation of such instruments.
Response 2: Thank you very much for your suggestions and related reminders regarding the appendices. We have considered your comments and decided to remove the questionnaire appendices from this study.
Round 2
Reviewer 3 Report
Comments and Suggestions for Authors
Thank you for your corrections.
Author Response
Comments1:I paid special attention to the blue highlights indicating recent revisions. The manuscript has been significantly improved in terms of content. A positive aspect is that the structure of the manuscript now flows logically and coherently. The problem statement and the relevance of the research are well-developed and firmly grounded in existing literature. The hypotheses are well-formulated, and the methodological justification is solid. However, there are still a few points that need attention. First, there are some grammatical inaccuracies in the English that should be corrected. For example, in the introduction on page 1, line 40, the sentence reads: “rigid behavioral patterns and rigid behavioral patterns.” This is redundant. Furthermore, on line 358 it says, “the mediating effect of also accounted,” which is grammatically incorrect. Regarding content, there is an inconsistency that needs to be addressed. The results section reports that Hypothesis 2, which posits school connectedness as the sole mediator, is not fully supported, because the independent mediation through school connectedness is not significant. However, the discussion section then states that school connectedness does partially mediate the relationship between coping and mental health. Isn’t this a contradiction? Could the authors clarify this? It might be helpful to include a clarification in the discussion, stating that the mediation was only significant through the chain (in combination with interdependent self-construal), not as a separate path.
Response1:Based on your feedback, we have carefully reviewed and revised the sentences and punctuation throughout the entire manuscript. I sincerely apologize for the inconsistency between the results section and the discussion section that you pointed out. It was a writing error. The correct conclusion is that the mediating effect of school connectedness is significant (Hypothesis 2), while the mediating effect of interdependent self-construal is not significant (Hypothesis 3). The writing error occurred because, before submitting to this journal, we restructured the discussion order of mediating variables in the introduction section to present the chain mediation effect more clearly. We discussed the second mediating variable (school connectedness) before the first mediating variable (interdependent self-construal). Therefore, the hypothesis regarding the mediating role of school connectedness has been revised from the original Hypothesis 3 to Hypothesis 2. However, the modification was overlooked in the results presentation section, which caused some confusion. Please understand and forgive.
Supplement the academic editor’s comments regarding insufficient revisions made in the first round of manuscript modification.
Comments2:(â… )The abstract could benefit from including more information. I would suggest that the authors briefly add the main effect sizes and mediation paths to the abstract to give readers a clearer picture of the findings. (II) I noticed no power analysis is included. Even with a large sample size, this is essential; (â…¢)Regarding statistical analysis: the authors do not report effect sizes or explained variance, and it’s unclear whether regression assumptions were systematically tested. A visual model including beta coefficients would significantly improve the clarity of the results.
Response2:(â… ) In the abstract section, we add the main effect sizes and mediation paths to make the results more clearer. (II) In the statistical methods section, we estimated the sample size using Monte Carlo simulations to complement the power analysis. The results indicated that at least 453 participants are required to achieve a statistical power of 0.8. (â…¢) In the results section, we supplemented the proportion of variance in the dependent variable explained by the mediation model. The chain mediation models explains 11% of the variance in depression, and 12% of the variance in anxiety. The regression coefficients between variables are described in the results section, including the beta coefficients. For the sake of simplicity, the beta coefficients are omitted in the figure. At the same time, Table 2 presented during the first revision has been removed, as its content is sufficiently explained in the text description.
